# Crystallinity Effect on Electrical Properties of PEALD–HfO_2_ Thin Films Prepared by Different Substrate Temperatures

**DOI:** 10.3390/nano12213890

**Published:** 2022-11-03

**Authors:** Xiao-Ying Zhang, Jing Han, Duan-Chen Peng, Yu-Jiao Ruan, Wan-Yu Wu, Dong-Sing Wuu, Chien-Jung Huang, Shui-Yang Lien, Wen-Zhang Zhu

**Affiliations:** 1Xiamen Key Laboratory of Development and Application for Advanced Semiconductor Coating Technology, School of Opto-Electronic and Communication Engineering, Xiamen University of Technology, Xiamen 361024, China; 2National Measurement and Testing Center for Flat Panel Display Industry, Xiamen Institute of Measurement and Testing, Xiamen 361024, China; 3Department of Materials Science and Engineering, National United University, Miaoli 36063, Taiwan; 4Department of Applied Materials and Optoelectronic Engineering, National Chi Nan University, Nantou 54561, Taiwan; 5Department of Applied Physics, National University of Kaohsiung, Kaohsiung University Rd., Kaohsiung 81148, Taiwan; 6Department of Materials Science and Engineering, Da-Yeh University, Changhua 51591, Taiwan

**Keywords:** HfO_2_ films, crystalline behavior, electrical properties, substrate temperature

## Abstract

Hafnium oxide (HfO_2_) thin film has remarkable physical and chemical properties, which makes it useful for a variety of applications. In this work, HfO_2_ films were prepared on silicon through plasma enhanced atomic layer deposition (PEALD) at various substrate temperatures. The growth per cycle, structural, morphology and crystalline properties of HfO_2_ films were measured by spectroscopic ellipsometer, grazing-incidence X-ray diffraction (GIXRD), X-ray reflectivity (XRR), field-emission scanning electron microscopy, atomic force microscopy and x-ray photoelectron spectroscopy. The substrate temperature dependent electrical properties of PEALD–HfO_2_ films were obtained by capacitance–voltage and current–voltage measurements. GIXRD patterns and XRR investigations show that increasing the substrate temperature improved the crystallinity and density of HfO_2_ films. The crystallinity of HfO_2_ films has a major effect on electrical properties of the films. HfO_2_ thin film deposited at 300 °C possesses the highest dielectric constant and breakdown electric field.

## 1. Introduction

Hafnium oxide (HfO_2_) is a promising material for its unique properties, such as high dielectric constant (k), high breakdown electric field, large band gap, excellent surface passivation performance, good stability, high refractive index and wide range of ultraviolet–infrared transparency region [1,2]. As a consequence of these properties, HfO_2_ film has captivated a tremendous amount of research interest for its applications in a variety of fields, such as anti-reflection films for ultraviolet lasers [3,4], high-k material in capacitors [5,6], non-volatile memories [7] and gate oxide in MOSFETs [8]. HfO_2_ film has wide applicability to both electronics and optoelectronics. It is used as a high-k material tunnel and gate oxide in nanocrystal floating gate non-volatile memories [9]. HfO_2_ film is further developed as a ferroelectric material for enhancing the memory windows [2]. HfO_2_ film also showed great applicability in short-wave infrared photosensors [10]. Furthermore, HfO_2_ film is utilized as an insulating material in perovskite solar cells [11], Cu(In,Ga)Se_2_ solar cells [12], c-silicon Passivated Emitter Rear Cell [13] and Polymer solar cells [14]. Due to its multiple applications, great efforts have been devoted to prepare and characterize the HfO_2_ films. In addition, the performance of HfO_2_ films is highly determined by the preparation methods. To obtain proper application, preparation of HfO_2_ film has been investigated through several methods, including ion beam sputtering [15], magnetron sputtering [16,17], molecular beam epitaxy (MBE) [18], metal organic chemical vapor deposition (MOCVD) [19,20], pulsed laser deposition (PLD) [21,22] and atomic layer deposition (ALD). HfO_2_ films obtained by MBE, MOCVD and PLD need to be prepared at a relatively high substrate temperature. Among these methods, ALD is considered to be one of most hopeful methods owing to its versatile advantages, such as accurate film thickness control, reproducibility, good conformity and high uniformity. ALD is able to meet the needs for atomic layer control and conformal deposition using a sequential, self-limiting surface reaction [23]. ALD is a thin film deposition technique based on the cycle-wise and alternate pulsing of precursor and reactant gases to a reactive surface [24]. ALD can be categorized into thermal ALD and plasma-enhanced ALD (PEALD). Compared with thermal ALD, PEALD offers many advantages, including a lower deposition temperature, a wider choice of feasible precursors and materials, advanced materials properties, more thorough surface ligand removal/regeneration and more flexibility for process optimizations [25]. The improved material properties are a result of the high reactivity provided by the plasma [26]. In PEALD, the surface is exposed to the species generated by a plasma during the reactant step. Typical plasmas used during PEALD are those generated in O_2_, N_2_ and H_2_ reactant gases or combinations thereof [27]. Such plasmas can replace ligand-exchange reactions typical of H_2_O or NH_3_ and can be employed to deposit metal oxides, metal nitrides and metal films. The substrate temperature is a key parameter that influences the property of HfO_2_ films grown by ALD. It determines the surface reaction and transformation of film structure. Therefore, substrate temperature has a critical effect on the crystalline behavior of the HfO_2_ film and, further, strongly affects the morphological and electrical properties of HfO_2_ films. Both thermal ALD and PEALD are successfully utilized to deposit HfO_2_ films. The growth of HfO_2_ films at various growth temperatures has been investigated by many research groups. In 2015, J. Gao et al. [28] reported that the grown temperature of HfO_2_ film prepared by thermal ALD must be kept at 200–240 °C to acquire a stable deposition rate of about 1 Å/cycle and the films with higher electric constant. Sai Li et al. [29] investigated structural and optical qualities of HfO_2_ films through ALD by adjusting the substrate temperature between 170 °C and 290 °C. D. Blaschke et al. [30] studied the hydrogen impurity level in thermal ALD deposited HfO_2_ films using tetrakis(dimethylamino)hafnium (TDMAHf) precursor and water at the growth temperature between 100 °C and 350 °C. In 2021, Matin Forouzmehr et al. [31] deposited HfO_2_ films on flexible polymeric substrates at temperatures changing from 100 to 250 °C. Although a lot of effort has been carried out to investigate the stability and interface chemistry of HfO_2_ films at a variety of temperatures, there exists much less literature about the effects of crystalline behavior on the electrical properties of HfO_2_ films. Additionally, there is an inalienable relationship between the electrical properties of HfO_2_ films and substrate temperature. Therefore, the crystalline structure affecting the electrical performance of HfO_2_ films deposited by PEALD was worthy of further investigation.

In this work, HfO_2_ films were grown by PEALD on silicon (Si) substrates. The substrate temperature was changed from 100 °C to 450 °C. The substrate temperature on the growth rate, surface morphology, crystalline behavior and electrical properties of the grown HfO_2_ films was comprehensively studied.

## 2. Experimental Methods

In this work, 4-inch p type Si wafers with a resistivity of 1–3 Ω·m were utilized as deposition substrates. Si substrates were ultrasonically cleaned by deionized water (10 s), 2% diluted hydrofluoric acid solution (1 min) and deionized water (10 s), respectively. After cleaning, the Si substrates were blown by nitrogen (N_2_) and transferred to the substrate holder. HfO_2_ films were prepared on Si at a temperature of 100, 200, 300, 400 and 450 °C using tetrakis (ethylemethylamino) hafnium (TEMAH, purity: 99.9999%, Aimou Yuan, Nanjing, China) and oxygen/argon (O_2_/Ar) plasma in a PEALD system (Picosun R-200, Espoo, Finland). The plasmas with the mixture of O_2_ and Ar gases were produced in a microwave cavity by an inductive coupling of radio frequency (RF) power (Litmas RPS, Advanced Energy, Denver, CO, USA). The plasma power was 2500 W. TEMAH was stored at a temperature of 120 °C in a bubbler-type stainless canister. N_2_ gas was utilized as the carrier gas for TEMAH. Its flow rate is 50 standard cubic centimeters per minute (sccm). The flow rate was controlled by a mass flow controller. The gas lines were heated to 130 °C, higher than the temperature of the bubbler-type stainless canister, to avoid the condensation of precursor. The base pressure of the reactive chamber was 100 Pa. The deposition process of PEALD HfO_2_ films sequentially included: TEMAH pulse time (1.6 s), N_2_ purge time (10 s), O_2_/Ar plasma processing (10 s) and N_2_ purge time (12 s). The deposition parameters of HfO_2_ films are listed in Table 1.

The thicknesses of HfO_2_ films were measured using spectroscopic ellipsometer (SE, 800 DUV, SENTECH, Berlin, Germany). The crystallinity and mass density of HfO_2_ films were measured through grazing incidence X-ray diffraction (GIXRD) and X-ray reflectivity (XRR). The diffraction pattern was acquired with an X-ray diffractometer through Cu K*α* irradiation with an incident angle of 1°. The initial scan was performed with a 2θ range of 10° to 80° at 0.02° step size and 4 min counting time. The XRR analysis was performed by a diffractometer using parallel beam geometry at an angle of incidence of 0° to 5°. The data analysis was carried out by the software SmartLab Studio II. The fitting model included a HfO_2_/SiO_2_/Si stack. The field emission scanning electron microscopy (FESEM, sigma 500, Oberkochen, Germany) and atomic force microscopy (AFM, XE7, Suwon, South Korea) characterizations were carried out to study surface morphology and roughness of HfO_2_ films. The chemical composition and bonding state in HfO_2_ films were measured by X-ray photoelectron spectroscopic analysis (XPS, ESCALAB 250Xi, Thermo Fisher, Waltham, MA, USA). Monochromatic Al Kα was used as an X-ray source. The X-ray spot size of 400 μm was detected for HfO_2_ films analysis. The binding energies were calibrated by a reference of C1s at the peak of 284.8 eV. The surface of HfO_2_ films was etched through Ar ion beam for 30 s to remove contaminants. The capacitance-voltage (C-V) and current-voltage (I-V) characterizations of HfO_2_-based devices were performed through a semiconductor parameter measurement system (Keithley 4200-SCS) with an EZON Probe Station at room temperature. For the device’s fabrication, Al/HfO_2_/Si capacitors were obtained through evaporating circular Aluminum (Al) dots with a metal mask (diameter: ~880 μm). Back contacts were also evaporating Al in a thermal system. Before real measurements, open circuit and short circuit calibration were performed. All the electrical tests were performed in a dark box.

## 3. Results and Discussion

To identify the growth per cycle (GPC) of HfO_2_ film, the HfO_2_ film thicknesses were characterized by SE at different positions. Thickness measurements were carried out after 100, 200, 300, 400 and 500 reaction cycles. Figure 1a presents the thickness of HfO_2_ films grown at 100, 200, 300, 400 and 450 °C as a function of the number of PEALD cycles. The thickness of HfO_2_ films increased linearly as the number of PEALD reaction cycles increased. This phenomenon indicated that the GPC is almost constant at each cycle. GPC can be determined from the slope of the linear fitting. Figure 1b illustrates the GPC of HfO_2_ thin films as a function of different temperatures. The GPC variation can be divided into three regimes. The GPC is approximately 0.13 nm/cycle at a substrate temperature of 100 °C. The GPC decreases to around 0.09 nm/cycle when the substrate temperature was increased to 200 °C. As the substrate temperature increases to 300 °C, the GPC of HfO_2_ film remains almost constant. Hence, a self-limited window temperature is found at 200–300 °C. When the substrate temperature was increased to 450 °C, the GPC experiences a big jump and reaches a value of 0.18 nm/cycle. The higher GPC at a temperature of 100 °C may result from the precursor of TEMAH condensation or physical adsorption on the surface of the Si substrate under lower temperature [28]. Therefore, when the HfO_2_ film is deposited at 100 °C, the surface reaction might deviate from the ideal ALD process, resulting in a higher GPC. In this case, the HfO_2_ films were deposited with a relatively loose structure. When the substrate temperature is varied from 200 °C to 300 °C, the GPC has almost no change, nearly 0.09 nm/cycle, indicating that the GPC of the HfO_2_ film is considerably stable. This phenomenon, generally called an ALD window, suggests a self-limiting surface reaction behavior. When the substrate temperature goes up to 400 °C and above, the precursor thermally decomposes. This behavior is similar to conventional chemical vapor deposition (CVD). The parasitic CVD-like processes during deposition result in a higher GPC. The CVD-like reaction in depositing HfO_2_ films at higher temperature was also found by In-Sung PARK et al. [32]. The precursor decomposition lies on the reactor design and process conditions. In this study, it was found that the CVD-like contribution was enhanced at high temperatures. In Ke Xu et al.’s study [33], a surge in the GPC of HfO_2_ film prepared by thermal ALD occurred beyond 250 °C, which resulted from precursor decomposition. However, according to Ke Xu et al.’s research, when the growth temperature was further increased to 275 °C, GPC decreased conversely, owing to predominant premature thermal decomposition of the precursor.

Figure 2a presents GIXRD patterns of HfO_2_ films deposited at different substrate temperatures. The GIXRD mode is applied, owing to the thin width of HfO_2_ films. When the substrate temperature is lower, such as 100 °C and 200 °C, HfO_2_ films show a broad feature at approximately 2θ of 32°, suggesting that the HfO_2_ films are amorphous. At lower temperature, the TEMAH and O_2_/Ar plasma do not have sufficient energy to migrate to the favorable sites. Additionally, the high concentration of physically absorbed precursors may also play a crucial role in generating the amorphous phase, as they may act as a steric hindrance and prevent chemically adsorbed precursors from migrating. When substrate temperature is increased, the precursors obtain more energy and can migrate to the energetically favorable sites. Meanwhile, the steric hindrance deriving from physically adsorbed precursors decreases. As the substrate temperature is further enhanced to 300 °C, some weak diffraction peaks present and the onset of crystallization occurs. Therefore, the HfO_2_ film transforms from amorphous to crystalline when the substrate temperature is enhanced from 200 °C to 300 °C. The phase transformation from amorphous to crystalline has also been reported by other groups [34]. When substrate temperature is higher than 300 °C, such as 400 °C and 450 °C, HfO_2_ films display many intense peaks, with different phase orientations which indicate an increase in the degree of crystallization. These peaks are indexed for HfO_2_ films and are compared with the standard JCPDS (data No. 06-0318) file. It is worth noting that the strongest (−111) phase of HfO_2_ film deposited at 400 °C transfers to the (200) phase when substrate temperature is increased to 450 °C. The transformation of the strongest phase in HfO_2_ films may be ascribed to the change in the stress of the films [35]. According to the research by Catalin Palade et al. [36], from HRTEM analysis of the HfO_2_ films prepared by magnetron sputtering and subjected to rapid thermal annealing, the stress field partially remaining in the lattice induces deformation of monoclinic and tetragonal phases after crystallization. The tetragonal structure formed in the crystal growth process changes into orthorhombic or monoclinic structures by a martensitic-like transition, depending on the doping and local stress field condition. The HfO_2_ region doped with Ge stabilizes the orthorhombic phase after martensitic-like transformation. Therefore, a transition of the lattice inside crystallites from the monoclinic to orthorhombic phase occurred, accompanied by a continuous strain deformation.

The sizes of HfO_2_ crystallites are obtained according to Scherrer’s formula (1), calculated through the intense (−111) peak data.
(1)d=kλ/(βcos θ)
where d, k, λ, *β* and θ refer to the sizes of HfO_2_ crystallites, the shape factor, the wavelength of X-rays, full width at half maximum (FWHM) and the Bragg’s angle, respectively. The FWHM of the (−111) peak increase is the increasing substrate temperature. The calculated average sizes of HfO_2_ crystallites decrease accordingly. The sizes of HfO_2_ crystallites grown at substrate temperatures of 400 °C and 450 °C are approximately 5.1 nm and 4.2 nm, respectively. However, the FWHM of (200) peak decrease is the increasing substrate temperature. Therefore, the calculated sizes of HfO_2_ crystallites increases, from 5.1 nm at a substrate temperature of 400 °C to 5.2 nm at a substrate temperature of 450 °C.

The increase in crystallization of HfO_2_ thin films with an increase in substrate temperature also affects the density of ALD–HfO_2_ films. Figure 2b presents the XRR pattern of HfO_2_ films deposited at various temperatures. The density of HfO_2_ films was extracted from the fitted XRR data. Figure 2c presents the density of HfO_2_ films as a function of substrate temperature. The density of the HfO_2_ films deposited at 100 °C is about 9.38 g/cm^3^. When the substrate temperature is raised from 100 °C to 300 °C, the density of HfO_2_ film increases and approaches the value of approximately 10.22 g/cm^3^. When the substrate temperature is further increased, the density of the films decreases. Usually, the density of the film depends on the chemical composition and crystallinity. The lower density of HfO_2_ films may be due to the insufficient surface reaction at a lower substrate temperature [37]. When the substrate temperature is lower, such as 100 °C, more impurities may be involved in the HfO_2_ film, which possibly derive from the dissociation of precursors. When the substrate temperature is enhanced from 100 °C to 300 °C, impurities in HfO_2_ films decrease [38]. Additionally, the crystallinity of HfO_2_ films increases. These two factors cause the improvement in the density of HfO_2_ films. When the substrate temperature is further enhanced to a higher temperature, the density of HfO_2_ films deteriorates due to the grain boundary and CVD-like reaction. The highest density of 10.22 g/cm^3^ obtained in this work is comparable to the reported density of HfO_2_ film using a MAP-Hf01 precursor and Ar/O_2_ plasma by Ji-hoon Baaek et al. [39]. The thicknesses of HfO_2_ films prepared 500 reaction cycles extracted from XRR are 64.7 nm, 46.4 nm,42.2 nm, 66.3 nm and 87.5 nm, respectively.

The surface morphology of HfO_2_ films at various substrate temperatures was measured by FESEM. The results are displayed in Figure 3. Figure 3a–e presents SEM images of HfO_2_ films prepared at a temperature of 100, 200, 300, 400 and 450 °C. All the figures have the same amplification factor to obtain a fair comparison. When the substrate temperature is lower, such as 100 °C and 200 °C, the HfO_2_ films exhibit a smooth surface, indicating a typical amorphous morphology. When the substrate temperature is 300 °C, some grains appear on the surface of the HfO_2_ film. When the substrate temperature is further raised to a higher value of 400 °C and 450 °C, larger grains with visible grain boundaries are present. Some larger grains might be the clusters consisting of smaller crystallites. The cluster-like surface morphology agrees well with the assumption of thermal decomposition of the TEMAH precursor and the CVD-like grown mode. To further characterize the surface morphological information, AFM measurements were carried out on the surface of the HfO_2_ films. The AFM images are inset on the top-right corners of the FESEM images. The root mean square (RMS) values obtained from a scan area of 1 μm × 1 μm in tapping mode are plotted in Figure 3f. The RMS value for HfO_2_ films grown at a substrate temperature of 100 °C is about 0.2 nm. When the substrate temperature rises to 200 °C, the RMS value sees a small increase. However, when the substrate temperature increases to 300 °C, the RMS value experiences a big jump, with a value of approximately 1.4 nm. The roughness of the HfO_2_ film rapidly increases due to the formation of crystallite [40]. When the substrate temperature further increases up to 400 °C, the RMS value maintains a small increase. When the substrate temperature rises to 450 °C, the RMS value again increases considerably. The surface of HfO_2_ films becomes rough as the substrate temperature is enhanced from 300 °C to 450 °C. This is attributed to the larger crystal grains embedded in HfO_2_ films. The AFM characterization of HfO_2_ films is in good agreement with GIXRD and FESEM measurements of HfO_2_ films.

The XPS characterizations were carried out to study the chemical composition, valence state of Hf, O and relevant defect sites in HfO_2_ films. Figure 4a presents surveyed XPS spectra of HfO_2_ films with a binding energy between 1200 and 0 eV. The peaks of Hf and O were investigated in detail. The photoemission intensity peaks Hf 4p1, Hf 4p3, Hf 4d3, Hf 4d5, Hf 4f and O 1s were detected by survey scan, along with auger electron peaks of O KLL. No other contamination species, except carbon, was found within the sensitivity of the instrument. The substrate temperature dependence of the atomic ratio of HfO_2_ films is plotted in Figure 4b. In all HfO_2_ films, the Hf atomic ratio was lower than 30% and the O-atomic-ratio was about 65% when compared with the stoichiometric HfO_2_. The reason for this lies in the carbon content decreasing the Hf atomic ratio. When the substrate temperature was raised from 100 to 450 °C, the O atomic ratio in the HfO_2_ film is almost constant, while the Hf atomic ratio in HfO_2_ film increases slightly. There are some carbon (C) impurities in HfO_2_ films. The carbon atomic ratio in HfO_2_ film has a decreasing trend except for the film deposited at a temperature of 450 °C. The C 1s atomic ratios in HfO_2_ films are 7.61%, 6.03%, 4.72%, 3.27% and 3.77%, respectively. The higher carbon concentration at lower substrate temperatures may be ascribed to the incomplete chemical reaction and amorphous structure of HfO_2_ films. According to the GIXRD measurement, HfO_2_ films were amorphous when the substrate temperature was lower than 300 °C. In this condition, carbon is more easily adsorbed in HfO_2_ films. The Hf 4f and O 1s peaks were analyzed in detail. The Hf 4f and O 1s peaks were fitted using the program XPSPEAK4.1 with a Gaussian–Lorentzian mixed function to confirm the chemical composition of HfO_2_ films. Figure 4c presents high-resolution spectra of Hf 4f of HfO_2_ films at various substrate temperatures. The Hf 4f spectra can be de-convoluted into two sets of double-peak components. One set of double-peak components at a binding energy of 17.2 eV and 18.8 eV is assigned to Hf^4+^ 4f_7/2_ and Hf^4+^ 4f_5/2_ peaks of the Hf oxide bond (O-Hf-O), respectively. Another set of double-peak components at a binding energy of 16.5 eV and 18.2 eV is assigned to Hf^x+^ 4f_7/2_ (x < 4) and Hf^x+^ 4f_5/2_ peaks of Hf suboxide bond, respectively [41]. The double-peak components of oxidized Hf^4+^ are stronger than those of the sub-oxidized Hf^x+^. The substrate temperature dependence of the area ratio of Hf^x+^/(Hf^x+^ + Hf^4+^) is plotted in Figure 4d. As the substrate temperature increases from 100 °C to 200 °C, the content of sub-oxidized Hf^x+^ for the HfO_2_ film decreases sharply. The content of sub-oxidized Hf^x+^ for the film remains nearly constant when the substrate temperature is between 200 °C and 300 °C. When the substrate temperature further rises up to 400 °C, the area ratio of Hf^x+^/(Hf^x+^ + Hf^4+^) increases significantly again. The higher content of Hf^x+^ for the film deposited at 100 °C could be ascribed to the relative shortage of the chemically absorbed precursors owing to the low energy at a low substrate temperature. Many defects, such as carbon and hydroxyl groups, are in the film, while the higher content of Hf^x+^ for the films deposited at 400 °C and 450 °C may be ascribed to thermal decomposition of the TEMAH precursor at high substrate temperatures. Figure 4e presents the high-resolution spectra of O 1s of HfO_2_ films at a variety of substrate temperatures. The peaks of O 1s for HfO_2_ films grown at the lower temperature of 100 °C and higher temperatures of 400–450 °C shift to higher binding energy. This phenomenon could be ascribed to more defects existing in the prepared films. The O 1s spectra can be de-convoluted into two components. The two components at the binding energy of around 530.5 eV and 531.5 eV indicate lattice oxygen (O_L_) and non-lattice oxygen (O_NL_), respectively. The O_L_ demonstrates that the prepared films have ordered structures with good properties. The O_NL_ demonstrates that non-lattice oxygen exists in the prepared films, which may originate from the suboxides with Hf [41], O-H or absorbed water. In order to investigate the amount of oxygen vacancies in HfO_2_ film, the area ratio of O_NL_/(O_NL_ + O_L_) as a function of substrate temperature is plotted in Figure 4f. The variation in the area ratio of O_NL_/(O_NL_ + O_L_) is similar to the variation in the area ratio of Hf^x+^/(Hf^x+^ + Hf^4+^). When the substrate temperature rises from 100 °C to 300 °C, the area ratio of O_NL_/(O_NL_ + O_L_) continuously decreases, indicating more perfect HfO_2_ lattice. This behavior is caused by the fact that the ratio of chemically adsorbed precursors increases and, therefore, more oxygen atoms can participate in forming Hf-O bonds owing to the increasing reaction energy. When the substrate temperature further increases to 400 °C and 450 °C, the area ratio of O_NL_/(O_NL_ + O_L_) increases to higher values of 28.6% and 28.3%. This behavior is possibly attributed to the decomposition of precursors at higher temperature.

C-V measurements are usually utilized to study the electrical property of dielectric films. C-V measurements were performed herein at room temperature, with the frequency of 1MHz on a standard Al/HfO_2_/Si/Al structure. Figure 5a illustrates normalized C-V characteristics of HfO_2_ thin films with various substrate temperatures. The applied voltage (V_A_) was varied (−6 V < V_A_ < 6V), with a sweep step length of 0.1 V, varying from accumulation to inversion. The variation in C-V curves toward negative voltages indicates the presence of effective oxide charges with positive polarity in prepared HfO_2_ films. Figure 5b presents the substrate temperature dependence of k of HfO_2_ films. The extracted k is identified by the maximum capacitance in the accumulation region. The k value is estimated by the following equation
$$k=\frac{C_{acc}}{A}\cdot \frac{d}{\varepsilon_{0}}$$
where $C_{acc}$ is the maximum capacitance in the accumulation region, A is the area of the capacitor, d is the thickness of the HfO_2_ film and $\varepsilon_{0}$ is the dielectric constant of the vacuum. The thickness of HfO_2_ films used for extracting the dielectric constant is approximately 30 nm. As can be seen in Figure 5b, the k value of the HfO_2_ film increases as the substrate temperature increases from 100 °C to 300 °C. According to the GIXRD measurement, the HfO_2_ film is mostly amorphous. The lower k value in the lower temperature range is due to the weaker energy of the precursor’s reaction at a lower substrate temperature, resulting in more defects in the prepared film. As the substrate temperature further increases from 300 °C to 450 °C, the k value of the HfO_2_ film decreases. According to the GIXRD measurement, the HfO_2_ film’s crystallization becomes obvious when the substrate temperature rises from 300 °C to 450 °C. The defects in crystallized HfO_2_ films are easily segregated at grain boundaries with unstable bonding, generating a leakage current path, leading to a decrease in k value [42]. Additionally, the decrease in k value in higher substrate temperatures may be ascribed to poor interface property, as presented in C-V curves. The lower slope of C-V curves for the capacitors of HfO_2_ film prepared at 400 °C and 450 °C indicates that more defects were presented in the interface of HfO_2_ films and Si substrates.

I-V measurements were performed to explore the leakage current properties of Al/HfO_2_/Si devices. Figure 5c presents I-V curves of the devices at different substrate temperatures. The corresponding breakdown electric field as a function of substrate temperature is also plotted in Figure 5d. As can be seen in the figure, the breakdown electric field firstly increases and then decreases as the substrate temperature increases. When the substrate temperature was 300 °C, the devices obtained the highest breakdown electric field, with the value of 5.88 MV/cm. The decreasing value of the breakdown electric field for the devices prepared at the temperatures higher than 300 °C may be ascribed to the highly crystallized HfO_2_ films that have a percolation path under the high electrical field due to the high density of the grain boundary defects [39]. The variation in the breakdown electric field is similar to the variation in k. This behavior is caused by the approximate breakdown electric field, E_bd_ ~ (k)^−1/2^ relation, illustrated by J. McPherson et al. [43].

Table 2 compares the qualities of HfO_2_ films obtained by different methods. All of the extracted k values of HfO_2_ films at various substrate temperatures agree with the reported ALD-grown HfO_2_ film. In this study, the highest obtained k value of 18.21 at the substrate temperature of 300 °C is higher than most of the other reported PEALD-grown HfO_2_ films [39,44,45], although it is slightly lower than the k value of 18.60 [46] prepared by PEALD with post deposition annealing and 18.3 [47] prepared by high power impulse magnetron sputtering (HIPIMS). The data of the breakdown electric field and density are not provided in references [46,47]. Therefore, further comparison is not possible. The k values prepared by Young Bum Yoo et al. [48] and Junhui Weng et al. [49] using spin-coating were 14.1 and 16.5, respectively. The k values obtained by Madhuchhanda Nath et al. [50] and A M Lepadatu [9] using radio frequency sputtering were 14 and 15, respectively. The k value obtained by Grzegorz Lupina et al. [51] is 16. The k values obtained by Devika Choudhury et al. [52] and Md. Mamunur Rahman et al. [53] using thermal ALD were 13 and 16.64, respectively. The highest breakdown electric field of the HfO_2_ films obtained in this work is 5.88 MV/cm, which is slightly lower than the value of 6.2 prepared by Young Bum Yoo et al. [48] using spin-coating. However, it is superior to most of the data provided by the comparative references. The highest density of 10.22 g/cm^3^ in this work is higher than other values of listing references. According to the investigation of the comparative table, the HfO_2_ film deposited in this study had a relatively higher mass density and lower defects density. Therefore, a high k value and breakdown electric field can be achieved.

## 4. Conclusions

In summary, HfO_2_ films have been successfully fabricated by PEALD at substrate temperatures ranging from 100 °C to 450 °C. The dependence of crystallization, film density and the electrical property of HfO_2_ films on substrate temperature has been systematically studied. GIXRD and XRR investigations revealed a gradual phase transformation from an amorphous structure to a polycrystalline structure and a higher crystallization at a substrate temperature of 400 °C. The substrate temperature has a major influence on the film density. XPS results indicate that HfO_2_ film grown at substrate temperatures of 300 °C has a sufficient chemical reaction, resulting in less Hf suboxide bonds and more lattice oxygen. Additionally, the phase transformation and the grain size of HfO_2_ films have a tremendous effect on the dielectric constant and breakdown electric field. HfO_2_ film prepared at 300 °C possesses the highest k value of 18.21 and a breakdown electric field of 5.88 MV/cm.

## Figures and Tables

**Figure 1 nanomaterials-12-03890-f001:**
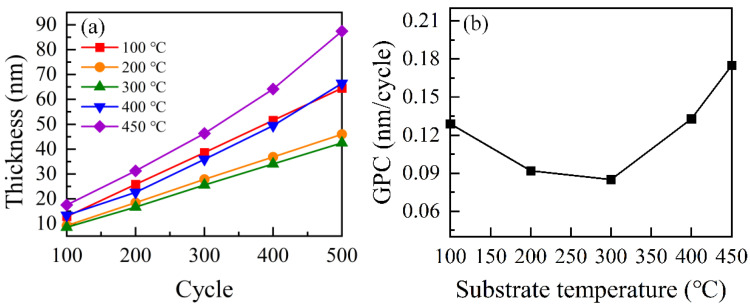
(**a**) The thickness of HfO_2_ film increased linearly with the number of ALD reaction cycles in PEALD. (**b**) GPC in a temperature range of 100–450 °C.

**Figure 2 nanomaterials-12-03890-f002:**
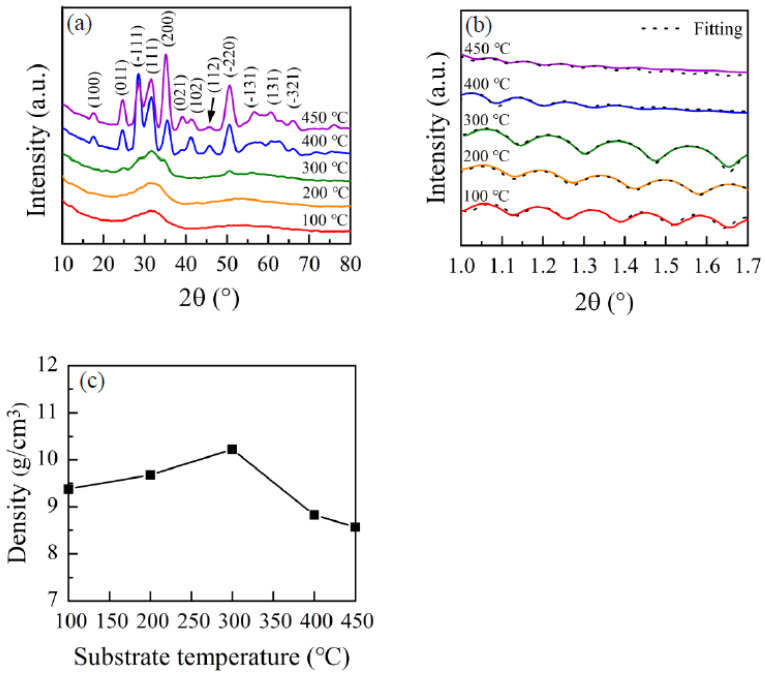
HfO_2_ films deposited at 100–450 °C: (**a**) GIXRD patterns; (**b**) XRR patterns and measured fitting curves; (**c**) the extracted density.

**Figure 3 nanomaterials-12-03890-f003:**
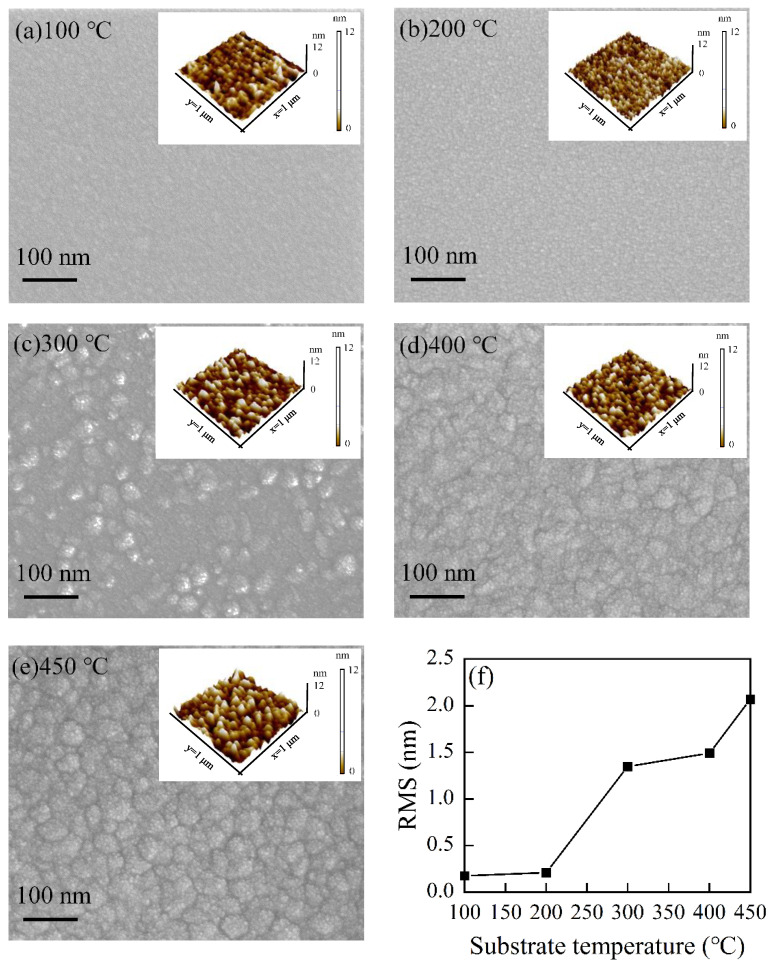
FESEM and AFM images of HfO_2_ films with (**a**) 100 °C, (**b**) 200 °C, (**c**) 300 °C, (**d**) 400 °C and (**e**) 450 °C. (**f**) RMS roughness of HfO_2_ by substrate temperatures.

**Figure 4 nanomaterials-12-03890-f004:**
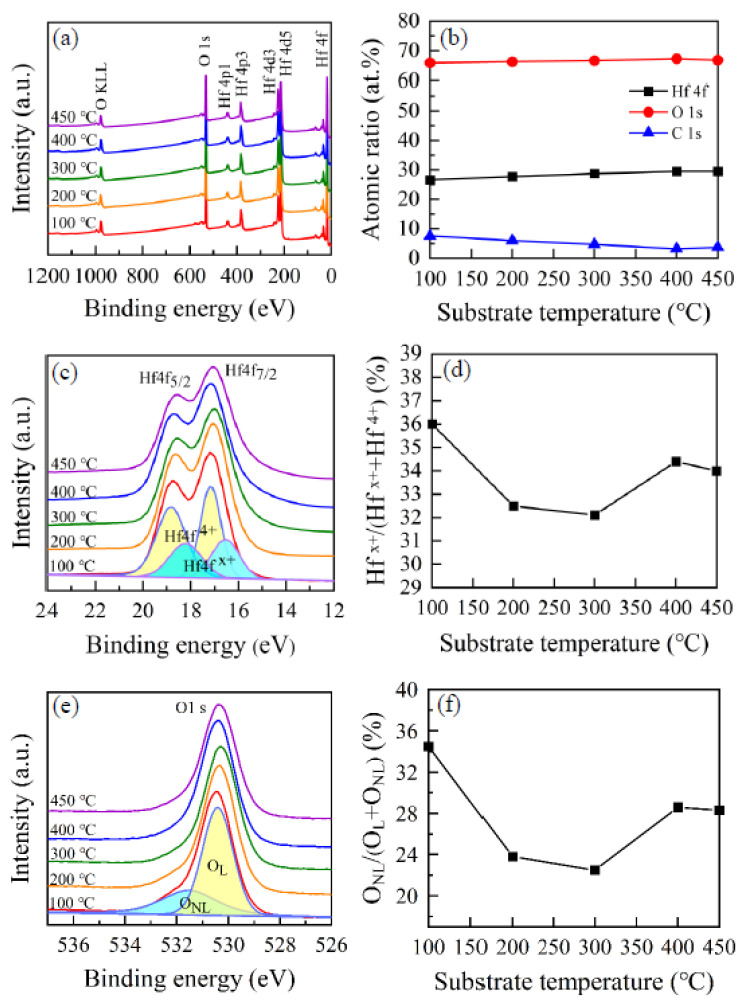
(**a**) XPS spectra of HfO_2_ films with a binding energy between 1200 and 0 eV. (**b**) Content proportion of Hf, O and C in HfO_2_ with different substrate temperatures. (**c**) High-resolution Hf 4f spectra of HfO_2_ films with various substrate temperatures. (**d**) The area ratio of Hf^x+^/(Hf^x+^ + Hf^4+^) as a function of substrate temperatures. (**e**) High-resolution O 1s spectra of HfO_2_ films with various substrate temperatures. (**f**) The area ratio of O_NL_/(O_NL_ + O_L_) as a function of substrate temperature.

**Figure 5 nanomaterials-12-03890-f005:**
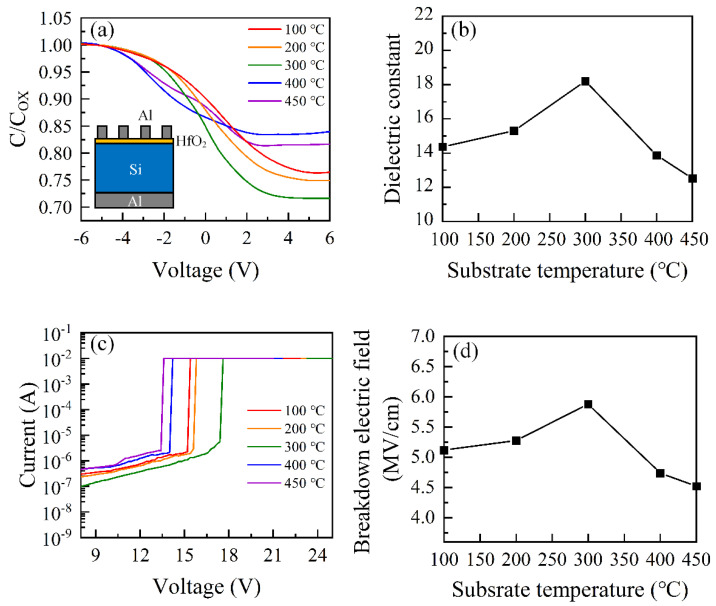
(**a**) C-V behavior of HfO_2_ MOS capacitors. (**b**) The extracted dielectric constant as a function of substrate temperature. (**c**) I-V curves of HfO_2_ MOS capacitors. (**d**) The breakdown electric field of the HfO_2_ MOS capacitor according to substrate temperature.

**Table 1 nanomaterials-12-03890-t001:** Deposition conditions of PEALD HfO_2_ thin films.

Parameter	Value
Bubbler temperature (°C)	120
TEMAH pulse time (s)	1.6
TEMAH purge time (s)	10
TEMAH carry gas flow rate (sccm)	50
O_2_ pulse time (s)	10
O_2_ purge time (s)	12
O_2_ flow stabilization (s)	2
O_2_ RF power on (s)	7
Ar flow rate (sccm)	50
O_2_ flow rate (sccm)	100
Substrate temperature (°C)	100–450
O_2_/Ar plasma Power (W)	2500

**Table 2 nanomaterials-12-03890-t002:** The Comparison of important parameters under different preparation methods.

Preparation Method	Film Thickness (nm)	Dielectric Constant	Breakdown Electric Field (MV/cm)	Density (g/cm^3^)	References
Spin-coating	100	14.1	6.2	7.8	[48]
Spin-coating	20	16.5	5.03	_	[49]
RF sputtering	5	14	_	_	[50]
RF sputtering	8	15	_	_	[9]
CVD	50	16	_	_	[51]
HIPIMS	_	18.3	_	_	[47]
thermal ALD	43	13	_	9	[52]
thermal ALD	5.1	16.64	4.8	_	[53]
PEALD	30	13.67	4.16	10.2	[39]
PEALD	4.7	16.64	_	_	[44]
PEALD PEALD	5.06 20	18.6 23	_ _	_ _	[46] [45]
PEALD	30	18.21	5.88	10.22	This work

## Data Availability

The data is available on reasonable request from the corresponding author.

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
