# Peer review of "Crystallinity Effect on Electrical Properties of PEALD–HfO2 Thin Films Prepared by Different Substrate Temperatures"

_nanomaterials, 2022, doi:10.3390/nano12213890_

Round 1
Reviewer 1 Report
The authors report on the electrical behavior function of crystallinity of PEALD HfO2 films for different substrate temperatures. They evidence the PEALD window, and the best temperature is 300 oC for which highest k value of 18.21 and breakdown electric field of 5.88 MV/cm were obtained as well as highest density of 10.22 g/cm3. The significance of obtained results from point of view of applications should be highlighted. The possible applications should be specified.
I have the following comments:
- Line 2 – Title: I suggest changing a little the title, example: Crystallinity effect on electrical properties ….
- Lines 37-42 - HfO2 applications: I want to highlight the high applicability of this material for both electronics and optoelectronics: HfO2 is used as high-k material tunnel and gate oxide in NCs floating gate NVMs [Nanotechnology 28, 175707 (2017), https://doi.org/10.1088/1361-6528/aa66b7], being further developed as ferroelectric material for enhancing the memory window [Nanotechnology 30, 445501 (2019), https://doi.org/10.1088/1361-6528/ab352b].
A lot of literature and patents focus on developing ferroelectric hafnia and devices based on them (see works of Uwe Schroeder and Thomas Mikolajick, Cheol Seong Hwang, e.g. [Nature Review Materials 7, 653 (2022), https://doi.org/10.1038/s41578-022-00431-2]).
HfO2 is much less used in optoelectronics, however it showed great applicability in SWIR photosensors as matrix oxide for embedding Ge-rich SiGe NCs and inducing strain at the same time with ensuring high quality NC/matrix interface leading to high photosensitivity in short-wave infrared to about 2000 nm exceeding Ge sensitivity limit [Materials 14, 7040 (2021), https://doi.org/10.3390/ma14227040].
- Line 83: Please specify the resistivity of Si wafers
- GI-XRD: What are the sizes of the HfO2 crystallites?
- Line 207 – Figure 2a GI-XRD: There seems to be a small tetragonal T contribution (~30 deg) corresponding to (101), if it is, please provide T crystalline fraction and monoclinic M crystalline fraction. Is there only M phase? Please provide fits of diffractograms from 27 to 33 deg (Lorentz peaks).
- Lines 183-186: “It’s worth noting that the strongest (-111) phase of HfO2 film deposited at 400 °C transfers to (200) phase when substrate 184 temperature is increased to 450 °C. The transformation of strongest phase in HfO2 films may be ascribed to the change of the stress of the films”. Such behavior was also reported in [J. Mater. Chem. C 9, 12353–12366 (2021), https://doi.org/10.1039/D1TC02921E]. Is there any variation of the main 2θ positions (possibly also M fraction) with the variation of GI angle? Did you check any gradual deviation from bulk values of HfO2 lattice fringes (HRTEM) – this could suggest stressed M and T structures as in [J. Mater. Chem. C 9, 12353–12366 (2021)]. The stress field partially remaining in the lattice induces deformation of M & T phases after crystallization. Maybe measurements of lattice fringes and some statistics on them could be useful.
- Line 207 – XRR: please provide extracted HfO2 film thicknesses.
- Lines 198, 282: there are some mistakes
- Are there any Cl impurities in HfO2 films?
- Line 360 -Table 2: Dielectric constants obtained by RF sputtering, see [Nanotechnology 28, 175707 (2017)] that reported “HfO2 is of high quality, having a dielectric constant ε = 15”.
- Please specify the thicknesses of films used for extracting the dielectric constant

Reviewer 2 Report
The manuscript presents a thorough investigation of Plasma ALD deposited HfO2 films, paying a special attention on the effect of the crystal structure on the electrical properties such layers.
The text is well written and the illustrations are adequate.
It is a bit difficult to judge the novelty of the work. Indeed, there are already several publications discussing the properties of the ALD HfO2 films. I trust authors when they say that the crystallinity effect is not yet well documented. In this sense it might be interesting for the ALD and microelectronics communities.
I recommend to publish the article with very tiny modifications regarding more the style than the substance:
line 28: I would replace "enormous" by "major" or similar word
Lines 89, 98 101, 169: You are talking about O2 plasma while both Ar and O2 were presented in the gas mixture. Please change to O2/Ar plasma or explain that you name it O2 despite the fact that argon was also presented
Line 170: Please change "a lot" to "high concentration of" or "numerous" or something similar, more formal
Line 346: You should choose if you write all words giving HiPIMS abbreviation a first capital letter or all with the small letters, but do not mix these two styles.
Reviewer 3 Report
The study presents the evolution of HfO2 films properties prepared by PEALD as a function of temperature. The results are interesting for the ALD and the microelectronic community. In fact, the manuscript provides a better understanding the crystalline and electric properties of hafnium oxide layers when prepared at different temperature by PEALD. The paper presents the results in an understandable way and the figures are clearly depicting the results.
Although the work is interesting – some changes should be carried out and properly addressed prior to publication.
1) First, the authors should rephrase many sentences and correct many misspellings all over the manuscript. Many typos and grammatical errors are present in the paper. For example, in the abstract, the sentence “The deposition rate, structural,(…)qualities of HfO2 films were carried out” is not correct, and should be rewritten. A deposition rate or a structureis not a “quality”, but a “property”. Second, those properties are not “carried out”, they are analyzed or measured. This is just an example, but there are many misspells and errors in the manuscript. The authors should rewrite the paper with a correct English or use external services to correct it.
2) The authors present their work without a proper introduction, and many references are missing. The first sentence were they describe why HfO2 is a promising material has for example no references at all! They could cite for example the following reviews: Fina et al, ACS Appl. Electron. Mater. 2021, 3, 4, 1530–1549 ; Schroeder et al, Nature Reviews Materials volume 7, pages 653–669 (2022). Similarly, the ALD technique should be better presented than just with one sentence, and the basics of the technique and references should be given, for example George, Chem review 110(1):111-31 2010, Weber et al, Nanotechnology 26 (9), 094002 (2015). Again, the importance of the Plasma assistance benefits should also be explained, and details on the relations between plasma properties and cristallinitys/films properties, should be given , along with references (Kessels et al, The Journal of Physical Chemistry C 118 (16), 8702-8711 (2014) ; Profijt et al, Journal of Vacuum Science & Technology A 29, 050801 (2011).
3) In line 152, the authors describe the ALD window obtained represented in the Figure 1b. They write “”parasitic CVD processes during deposition at high temperature results in a higher growth rate”. This statement is true and has been described by others as well. When looking at Fig 2a and Fig 4b, however, one can observe that this higher temperature (around 400°C) leading to parasitic CVD, also leads to better crystallinity and lower C contamination. Th authors should explain why CVD would then lead to better material properties and why they don’t use it as a deposition technique instead of ALD.
4) In table 2, the authors compared the properties of hafnium oxide films prepared with different techniques. It is of critical importance to add the thickness of the films, as the properties depend on it. Also, films prepared by CVD technique should be added.
5) The term “quality” should be replaced with “property” all over the manuscript.
6) The main finding claimed in the paper is that the electrical properties of the films are related to their crystallinity. This is true but is not new; the authors should give additional details and extract more important findings from the results presented.
Reviewer 4 Report
This work is not innovative, there are many HfO2 papers by ALD and PEALD with this precursor
however, it is interesting to read, well written, and it can be useful.
some remarks:
- the ALD community uses the term GPC (Growth per cycle) rather than deposition rate; can you use this term in your figure 1b?
- Can you be more precise about the origin of the O1s peak at 531.5 eV described as oxygen deficiency? Can't you attribute it to e.g. O-H bonds or absorbed water?
- The dielectric constants are rather low for a crystalline HfO2. I see two reasons for this: your material is only crystallized in monoclonal form (which gives a low K value for HfO2), is this true? I didn't see any comment in your paper about the form but maybe I read it wrong. Secondly, it is very likely that you have a SiO2 or HfSiO2 type interface giving a low dielectric constant and thus reducing the overall measured value. Do you consider this interface in your calculation of k value from C(V) measurements?
Concerning the breakdown field, a further explanation is simply that the breakdown depends on the local field which depends on the value of k (cf Lorentz relation and all papers of the years 2000 like the one of J. McPherson et al, Appl. Phys. Lett. 82 (2003) 2121)
Round 2
Reviewer 3 Report
This updated version is fine to be published
Reviewer 4 Report
The authors have taken into account the remarks of the reviewers. I think the paper can be published as it is.